# Antimicrobial and Regenerative Effects of Placental Multipotent Mesenchymal Stromal Cell Secretome-Based Chitosan Gel on Infected Burns in Rats

**DOI:** 10.3390/ph14121263

**Published:** 2021-12-04

**Authors:** Vasily A. Kudinov, Rafael I. Artyushev, Irina M. Zurina, Roman D. Lapshin, Ludmila B. Snopova, Irina V. Mukhina, Olga S. Grinakovskaya, Irina N. Saburina

**Affiliations:** 1Laboratory of Cell Biology and Developmental Pathology, FSBSI Institute of General Pathology and Pathophysiology, 125315 Moscow, Russia; izurina@gmail.com (I.M.Z.); grinakovskaya@yandex.ru (O.S.G.); saburina@mail.ru (I.N.S.); 2Scientific Group of Phospholipid Drugs, Institute of Biomedical Chemistry, 119121 Moscow, Russia; rifiraf21ibmx@gmail.com; 3Institute for Regenerative Medicine, Sechenov First Moscow State Medical University, 119991 Moscow, Russia; 4Central Research Laboratory, Privolzhsky Research Medical University, 603005 Nizhny Novgorod, Russia; r.d.lapshin@gmail.com (R.D.L.); lsnopova@mail.ru (L.B.S.); mukhinaiv@mail.ru (I.V.M.)

**Keywords:** thermal burn, infection, wound healing, *Staphylococcus aureus*, chronic wound, exosome, cell-free therapy

## Abstract

Background: There is a need for better strategies to promote burn wound healing and prevent infection. The aim of our study was to develop an easy-to-use placental multipotent mesenchymal stromal cell (MMSC) secretome-based chitosan hydrogel (MSC-Ch-gel) and estimate its antimicrobial and regenerative activity in *Staphylococcus aureus*-infected burn wounds in rats. Methods: Proteomic studies of the MMSC secretome revealed proteins involved in regeneration, angiogenesis, and defence responses. The MMSC secretome was collected from cultured cells and mixed with water-soluble chitosan to prepare the placental MSC-Ch-gel, which was stored in liquid phase at 4 °C. The wounds of rats with established II-IIIa-degree burns were then infected with *S. aureus* and externally covered with the MSC-Ch-gel. Three additional rat groups were treated with medical Vaseline oil, the antiseptic drug Miramistin^®^, or the drug Bepanthen^®^ Plus. Skin wound samples were collected 4 and 8 days after burning for further microbiological and histological analysis. Blood samples were also collected for biochemical analysis. Results: Application of the MSC-Ch-gel cleared the wound of microorganisms (*S. aureus* wasn’t detected in the washings from the burned areas), decreased inflammation, enhanced re-epithelialisation, and promoted the formation of well-vascularised granulation tissue. Conclusions: MSC-Ch-gel effectively promotes infected wound healing in rats with third-degree burns. Gel preparation can be easily implemented into clinical practice.

## 1. Introduction

Every year, around the world, many patients suffer from burn wounds that do not completely heal or result in scarring and chronic wound formation [1]. Burn wounds represent susceptible sites for opportunistic colonisation by endogenous and exogenous organisms. Microbial infections are a leading cause of delayed wound healing, contributing to this chronic wound state [2]. The existing therapy for burn wounds is mainly focused on preventing them from becoming infected and the application of antiseptic and anti-inflammatory drugs. At the same time, the use of antibacterial drugs and antibiotics is associated with the development of side effects and the spread of resistant strains of microorganisms [3].

The wound repair process is complex and involves a set of events that include inflammation surrounding the region of injury, wound cell migration and mitosis, angiogenesis and the development of granulation tissue, and repair, regeneration and remodelling of the extracellular matrix (ECM) and connective tissue that ultimately heals the wound [4,5,6]. Colonisation of the wound with opportunistic and pathogenic microorganisms disrupts the healing process and can lead to the development of chronic wounds that threaten the patient’s life. Additionally, microorganisms can form bacterial films, mainly associated with granulation tissue (eschar) in the chronic wound bed. The most common bacteria implicated in chronic wound biofilms include *Staphylococcus aureus*, *Pseudomonas aeruginosa* and β-haemolytic Streptococci; other bacteria include *Enterococcus* spp., *Klebsiella pneumoniae*, *Acinetobacter baumanii* and *Enterobacter* spp., coagulase-negative *Staphylococci* and *Proteus* spp. [7]. In chronic wound granulation tissue, keratinocytes and fibroblasts possess reduced migratory and proliferative capacity, resulting in decreased production and dysregulated inflammatory and antimicrobial responses. Moreover, microbial infection stimulates massive influx of neutrophils and microphages in the wound however their phagocytic, chemotactic, and antimicrobial activity is decreased. Dysregulation of the immune response in the wound leads to an excessive increase in the activity of metalloproteinases, disruption of the ECM remodelling, decrease of neovascularisation and blood flow in the area, activation of free radical processes, hypoxia, shifting the chronic wound microenvironment to an alkaline pH, which not only has a detrimental effect on host cellular function but possibly promotes biofilm formation, thereby promoting the prolonged inflammatory-proliferative phase [6,8,9,10,11]. Multi-targeting treatments are required to exit out of this vicious cycle and achieve optimal and rapid recovery of tissue damage.

Recently, cellular therapies using multipotent mesenchymal stromal cells (MMSCs) have offered a new bio-therapeutic approach. MMSCs are self-renewing multipotent cells found in numerous locations within the body, such as bone marrow, adipose tissue, endometrium and placenta [7]. MMSC-based cell therapy has already been shown to be an effective approach in stimulating skin wound healing by exhibiting immunomodulatory and anti-fibrotic properties, promoting the migration of fibroblasts and keratinocytes, stimulating angiogenesis, and exerting bactericidal activity [12,13,14,15]. There are numerous encouraging preclinical studies using different MMSC therapy approaches for the treatment of acute skin burns, including systemic cell administration, local cell injections, and cell-seeded scaffolds [13]. MMSCs are also known to possess antimicrobial activity, which has prompted their study as therapeutic agents for infections [15]. MMSCs can realise their antimicrobial activity through direct mechanisms, via the secretion of antimicrobial peptides or via indirect mechanisms that regulate the host immune response. It was shown that MSC-derived antimicrobial peptides act directly on bacterial surface and intracellular molecular targets and include cathelicidin (LL-37), defensins and lipocalins [16,17,18]. The indirect antimicrobial effects of MMSCs, partly mediated via toll-like receptor (TLR) signalling, linked to modulation of proinflammatory cytokine and chemokine response, inhibition of excessive proliferation and infiltration of inflammatory T cells, natural killer cells, microphage repolarisation, and modulation of phagocytic activity of monocytes and neutrophils [7,16,19,20].

Many of these therapeutic effects of MMSCs are mediated by paracrine factors, also known as the secretome [21,22]. The secretome contains extracellular vesicles (EVs), growth factors, cytokines, and chemokines, and is regarded as a promising cell-free alternative therapeutic agent. MMSCs are the preferred source of secretome, since MMSCs themselves appear to be safe based on a large amount of clinical data generated over the last decade [23,24,25,26,27,28,29].

In this respect, cell-free therapies using secretory products derived from MMSC cultures (conditioned media or EVs) have recently become a research hotspot, including research in skin wound healing [30,31]. The use of these approaches possesses advantages such as lower risks of tumorigenesis, embolism, and pathogen transmission, as well as the capacity for long-term storage while preserving activity. Repeated subcutaneous injections of conditioned medium from rat bone marrow MMSC (BM MMSC) monolayer cultures after radiation-induced skin burns resulted in improved tissue regeneration due to reduced leukocyte infiltration in a rat model [32]. Multiple intraperitoneal injections of conditioned medium from BM MMSCs have also promoted skin regeneration after second-degree burns by reducing inflammatory cells and stimulating fibroblast proliferation, collagen synthesis, and angiogenesis [33]. In addition, the topical application of conditioned medium from BM MMSCs on full-thickness excisional wounds for four days has been shown to stimulate skin regeneration, and this effect could be significantly enhanced by culturing the BM MMSCs under hypoxic conditions [34]. Incubation of MMSCs with Nod-like receptor agonists, Toll-like receptor agonists and a key pro-inflammatory cytokine (interferon-gamma) may also elicit cell activation and enhance their bactericidal activity [20].

In most studies, MMSCs derived from bone marrow, Wharton’s jelly derived from the umbilical cord, and adipose tissue have been used as the sources of the secretome or EVs. The placenta is also an available post-partum source of MMSCs and it creates no ethical controversies for secretome production in clinical practice [35]. While the proteome profiles secreted by MMSCs from different sources cultured under the same conditions share similar functional features (mainly related to cell proliferation and migration, ECM synthesis, and anti-apoptosis), the proteomes of post-partum MMSCs (from the umbilical cord and placenta) have been predicted to have higher therapeutic potential [36]

In this study, we aimed to develop a placental MMSC secretome-containing chitosan gel for wound healing. Chitosan is a hydrophilic biopolymer obtained by alkaline deacetylation of chitin, a major component of arthropod shells [37]. Based on its mucoadhesion, biocompatibility, antimicrobial activity, biodegradability, unique physicochemical properties, and lack of toxicity, chitosan has received considerable attention for its pharmaceutical and medical applications including drug delivery, cell encapsulation, and wound dressing [38,39,40,41,42,43]. Chitosan can synergise with the secretome to accelerate burn recovery [44,45,46]. The hydrogel may also protect exosomes and proteins from proteases in the wound site and maintain the optimal concentration of MMSC-derived bioactive molecules and exosomes in the wound [4,47].

We have developed a simple and available method for preparing a formulation based on water-soluble chitosan and the secretome from placental MMSCs and have evaluated its biological activity in a model of *Staphylococcus aureus*-infected burns (II-IIIa degree) in rats. We hypothesise that that the combination of chitosan and the secretome from placental MMSCs would have optimal effect on the healing process, stimulating regeneration and protecting the wound from microorganisms. The antiseptic drugs benzyldimethyl[3-myristoylamine)-propyl]ammonium chloride monohydrate (Miramistin^®^) and Bepanthen^®^ Plus containing chlorhexidine hydrochloride and dexpanthenol, widely used in the clinic, were used as treatment controls [48,49].

## 2. Materials and Methods

### 2.1. Chemicals and Reagents

The following reagents were purchased for the study. Anti-CD105, anti-CD90, anti-CD73, anti-CD44, anti-CD29, anti-CD14, anti-CD31, and the mix of anti-CD45/anti-CD34/anti-CD11b/anti-CD19 were purchased from BD Bioscience (San Jose, CA, USA) for cell immunophenotyping. Hematoxylin and Eosin (ab245880, Abcam, Cambridge, UK) were purchased for histology. Trypsin solution (BioLot, Saint-Petersburg, Russia), Versene solution (BioLot), Hank’s Balanced Salt Solution (BioLot), Petri dishes (Corning-Costar, Tewksbury, MA, USA), Dulbecco’s Modified Eagle Medium/Nutrient Mixture F-12 (DMEM-F12; BioLot), glutamine (BioLot), gentamicin (PanEco, Moscow, Russia), insulin-transferrin-selenium (ITS-G; BioLot), basic fibroblast growth factor (bFGF; ProSpec, Ness-Ziona, Israel), heparin (PanEco), and foetal calf serum (FCS; HyClone, South Logan, UT, USA) were purchased for cell culturing. Phosphoric acid (Sigma, Hamburg, Germany), methanol (J.T.Baker, Arnhem, Netherlands), urea (Sigma, Germany), sodium chloride (Fluka-Honeywell, Seelze, Germany), sodium deoxycholate (Sigma, Milan, Italy), acetonitrile (Carlo Erba, Val de Reuil, France), triethylammonium bicarbonate (Sigma, Switzerland), tris(2-carboxyethyl)phosphine (TCEP; Sigma, St. Louis, MO, USA), 4-vinylpyridine (Aldrich, Gillingha, UK), isopropanol (Fisher Chemical, Loughborough, UK), trypsin (Promega, Madison, WI, USA), acetic acid (Carlo Erba, France), ethyl acetate (Carlo Erba), acetonitrile (Carlo Erba), formic acid (Sigma, Germany) were purchased for proteomic analysis of the cell secretome. Low molecular weight water-soluble chitosan (Bioprogress, Moscow, Russia) was procured for the gel preparation. Bepanthen^®^ Plus wound healing cream (chlorhexidine hydrochloride + dexpanthenol; Bayer Pharma AG, Grenzach-Wyhlen, Germany) and Miramistin 0.01% solution (benzyldimethyl[3-myristoylamine)-propyl]ammonium chloride monohydrate; INFAMED, Moscow, Russia) were used as positive controls in the animal studies.

### 2.2. Cell Sources

The study was conducted using the primary cultures of human placenta-derived MMSCs. Placenta tissues were collected from three donors (age 23–36) undergoing Caesarean sections at 39–41 weeks of gestation after receiving patients’ written, informed consent. All these procedures were carried out under aseptic conditions and were approved by The Local Ethical Committee of the FSBSI “Institute of General Pathology and Pathophysiology” and performed in accordance with the Helsinki Declaration.

### 2.3. Animals

Forty 3-month-old male Wistar rats were purchased from the vivarium of Privolzhsky Research Medical University (Nizhny Novgorod, Russia). The rats were individually housed in stainless steel cages in an air-conditioned room (17 ± 1 °C, 55 ± 5% humidity) with a 12 h/12 h light/dark cycle and ad libitum access to food and water. All experiments were conducted to minimise suffering and the number of animals used.

### 2.4. Bacterial Species

Clinical isolate of gram-positive *S. aureus* was used as the test organism in this study.

### 2.5. MMSC Cell Culturing and Secretome Collection

Placental tissue transported to the laboratory was first washed in Hank’s solution containing 100 U/mL gentamycin for 12 h and was then mechanically cut into fragments of no more than 3 mm. Individual cells were isolated from the placental tissue pieces by incubation for 30 min in 0.15% collagenase type II solution (Sigma, Germany) at 37 °C with constant stirring.

Hank’s solution was added to the isolated cells to reduce the collagenase activity. The resulting suspension was filtered through a 100-μm-pore filter (Becton Dickinson, Franklin Lakes, NJ, USA) and centrifuged at 300× *g* for 10 min. The cell pellet was resuspended in full growth medium and plated in Petri dishes at the density of 3 × 105 cells/cm^2^ under standard conditions (37 °C in 5% CO_2_). The full growth medium consisted of basal DMEM-F12 with 2 mM glutamine supplemented with 40 U/mL gentamicin, 1% 100 × ITS-G, 20 ng/mL bFGF, 15 U/mL heparin, and 10% FCS. The medium was changed 2–3 times per week, and visual inspection of the culture was performed under a Primovert phase-contrast microscope (Zeiss, Jena, Germany). The cell cultures were passaged at a 1:3 ratio upon reaching 80–85% confluence using Versene and 0.25% trypsin solution.

At passage three, when the placental MMSCs reached 80–85% confluence, the medium was replaced by a growth medium without gentamicin, and cells were cultured for 72 h without changing the medium. After 72 h, 500 µL of the resulting conditioned medium (CM) was collected individually from every dish for further content analysis. The remainder of the CM was pooled together for each cell culture in 50 mL tubes (10 mL of CM from 2 × 10^6^ cells), filtered through 0.22 µm filter to remove any cell and cell debris. The filtered secretome was frozen and stored at −80 °C.

### 2.6. Immunophenotyping of Human Placental MMSCs

Placental MMSCs were subjected to immunophenotyping after 3 days of passage using the following surface marker profile that is characteristic of MMSCs: CD105, CD90, CD73, CD44, CD29, CD31, CD14 and CD45, CD34, CD11b, CD19 together in a premixed form [50,51]. Specifically, the cell suspension obtained by treatment of the monolayer with Versene and 0.25% trypsin was transferred to 15 mL tubes and centrifuged (7 min, 400× *g*). The pellet was resuspended in phosphate-buffered saline (pH 7.4) containing 1% serum and incubated in the dark (15 min, 25 °C) with antibodies (10 μL of antibodies per 1 × 10^6^ cells) conjugated with fluorescent labels (fluorescein isothiocyanate (FITC), phycoerythrin (PE), Peridinin chlorophyll protein-Cyanine5.5 (PerCP-Cyanine5.5) and Allophycocyanin (APC)). The stained cells were centrifuged (5 min, 400× *g*), and the pellet was resuspended in 1 mL phosphate-buffered saline containing 1% FCS in tubes for flow cytometry. The samples were analysed on a Sony SH800 flow cytometer (Sony, Japan).

### 2.7. Proteomic Analysis of the Cell Secretome

#### 2.7.1. Sample Preparation

One hundred microlitres of each sample was transferred into clean tubes and 6 μL of 85% phosphoric acid (up to 5% final concentration; Sigma, Germany) was added and mixed. Then, 800 μL of methanol (J.T.Baker) was added and the samples were mixed again. The resulting suspension was centrifuged at 10,000× *g* at 15 °C for 10 min (Centrifuge 5424R, Eppendorf, Hamburg, Germany). The precipitate was reconstituted in 50 μL of a denaturing solution consisting of 5 M urea (Sigma, Germany), 1% sodium deoxycholate (Sigma, Italy), 300 mM sodium chloride (Fluka-Honeywell), 10% acetonitrile (Carlo Erbo), 100 mM triethylammonium bicarbonate (pH 8.2–8.5) (Sigma, Switzerland), and up to 10 mM freshly added neutralised TCEP (Sigma, St. Louis, MO, USA). The reconstituted denatured protein was incubated at 45 °C for 30 min with constant vigorous stirring at 1200 rpm (Thermo Mixer, Eppendorf, Germany). Then, 6 μL of 2% stabilised 4-vinylpyridine (Aldrich, UK) in 30% isopropanol (Fisher Chemical) was added to a final concentration of 0.2%. The alkylation reaction was incubated in the dark at 20 °C for 20 min. The sample volume was then increased to 500 μL by adding 384 μL of 75 mM triethylammonium bicarbonate (pH 8.2) and thoroughly mixed.

To each sample, 400 ng of trypsin was added from a 100 ng/μL stock (Promega) in 30 mM acetic acid (Carlo Erba) and incubated for 3 h at 40 °C with intermittent stirring (stirring at 1700 rpm for 90 s every 10 min). Then, an additional aliquot of 400 ng trypsin (100 ng/μL) in 30 mM acetic acid was added and the reaction was incubated for 2 h at 42 °C with intermittent stirring as above. At the end of the incubation, 10 μL of absolute formic acid was added to precipitate the reduced deoxycholic acid. The resulting suspension was centrifuged at 12,000× *g* and 5 °C for 10 min. To remove the residual deoxycholic acid, an equal volume of ethyl acetate (Carlo Erba) was added to 500 μL of the supernatant and vigorously stirred for 3 min at room temperature. Then, the mixture was centrifuged at 10,000× *g* for 5 min at 20 °C and then incubated at −20 °C for 10–15 min. The samples were removed from the freezer, the surface organic layer was decanted, and 150 μL of acetonitrile (Carlo Erba) was added to the lower aqueous layer containing the peptides. The mixture was centrifuged at 13,000× *g* for 10 min at 20 °C, and the supernatant was collected and dried under vacuum at 30 °C for 60–70 min with chamber ventilation every 15 min (Concentrator Plus, Eppendorf, Germany). The resulting dry residue was reconstituted in 20 μL of 0.5% formic acid (Sigma, Germany).

#### 2.7.2. High-Performance Liquid Chromatography-Mass Spectrometry (HPLC-MS)

The analysis was performed on a Xevo™ G2-XS QToF quadrupole time-of-flight mass spectrometer (Waters, UK) coupled to an Acquity™ HPLC H Class Plus chromatography system (Waters). The analysis was carried out in positive electrospray ionisation mode with increased sensitivity and a normal dynamic range of measurement. The emitter voltage was 3 kV, the drying gas rate was 680 L/min, the focusing gas rate was 50 L/min, the temperature of the ionisation source was 150 °C, and the temperature of the desolator was 350 °C. The voltage across the focusing cone was 67 V with a bias of up to 130 V. The ions were recorded in hybrid data-independent acquisition (DIA) MSe-SONAR mode. Specifically, an initial DIA MS scan was performed from 100–1500 *m/z*, followed by a SONAR scan with mass isolation with a quadrupole from 400–1100 *m/z* and an isolation peak width of 22 Da. The time for one complete scan cycle was set at 0.418 s. Fragmentation was carried out in two-phase mode: phase 1—low-energy collision-induced dissociation (CID) fragmentation with argon at 6 eV; phase 2—high-energy ranked CID fragmentation with argon from 15–40 eV. During the analysis, active mass correction (*m/z* = 556.27) with a low activation energy (9 eV) was performed using a leucine enkephalin standard (50 pg/mL in 50% acetonitrile with 0.1% formic acid). The standard was injected into the ionisation source for 30 ms at 5 μL/min every 45 s and isolation within 200 mDa.

Chromatographic separation was performed on an Acquity™ UPLC BEHC18 column (1.7 μm particle size, 2.1 × 50 mm column size; Waters) at a flow rate of 0.2–0.3 mL/min and constant temperature of 40 °C. Gradient elution was used, consisting of mobile phase A (aqueous solution of 0.1% formic acid and 0.03% acetic acid) and mobile phase B (solution of 0.1% formic acid and 0.03% acetic acid in acetonitrile) with the following elution scheme: 0–1.5 min, 3% B; 1.5–26.5 min, increased to 19% B; 26.5–42 min, increased to 32% B; 42–43.5 min, increased to 97% B; held in isocratic mode until 47.5 min; decreased to 3% B until 49 min; and held in isocratic mode until 53 min. The flow rate from 43.5–47.5 min was 0.3 mL/min; at all other times, the flow rate was 0.2 mL/min.

#### 2.7.3. Data Analysis

Raw data files were handled using the PLGS (Protein Lynx Global Server, version 3.0.3, Waters, the UK) search engine. Proteins search was performed against human amino acid sequences database obtained from the UniProt KB (release May 2021) as a FASTA file with automatically generated reversed concatenated decoy sequences to estimate a false positive rate. Precursor ion mass tolerance was set at 20 ppm (±10 ppm tolerance window) and fragment ion mass tolerance was set to 8 mDa (±4 mDa tolerance window). S-pyridilethylation was included as a fixed modification, whereas oxidised methionine (oxM), deamidated glutamine(dQ) and asparagine (dN) were variable modifications. The minimal peptide length was fixed to eight amino acids with only one allowed internal missed cleavage. The false discovery rate (FDR) at 1% was determined for peptide and protein identification by accumulating the reverse database hits.

The listed secreted proteins differing in the samples were classified using PANTHER (Protein Analysis THrough Evolutionary Relationships, http://pantherdb.org, accessed on 1 December 2021) to study molecular functions, cellular components, and pathways. For subcellular classification, the protein subcellular localisation predictor mGOASVM (human) was used [52].

### 2.8. Multiplex Detection Immunoassay of Secretome Proteins

Conditioned medium from the MMSC cultures was collected after 3 days and stored at −80 °C. The stored supernatants were thawed and centrifuged at 500× *g* for 3 min to eliminate cell debris and then analysed by magnetic bead-based Bio-Plex multiplex assays for cytokines, chemokines, and growth factors, including bFGF, interferon-gamma (IFN-γ), interleukin (IL)-12 (p70), IL-13, CCL5 (RANTES), CCL3 (MIP-1α), CCL4 (MIP-1β), vascular endothelial growth factor (VEGF), IL-1β, IL-2, IL-4, IL-5, IL-6, CCL2 (MCP-1), CCL11 (eotaxin-1), IL-8, IL-9, IL-10, IL-13, IL-15, IL-17A, IL-1RA, granulocyte-macrophage colony-stimulating factor (GM-CSF), granulocyte-colony stimulating factor (G-CSF), tumour necrosis factor-alpha (TNF-α), IL-7, and CXCL10 (IP-10) (Bio-Plex Pro™ Human Cytokine 27-plex, Bio-Rad). The assay was performed according to the manufacturer’s instructions using luminex xMAP (multi-analyte profiling) technology.

### 2.9. Preparation of the Chitosan Gel

To obtain a 2% (*w*/*v*) gel, a water-soluble low-molecular-weight chitosan (Bioprogress, Russia) was used. A weighed amount of chitosan was added to the MMSC conditioned medium and stirred slowly overnight at 4 °C. The resulting gel was stored at 4 °C for 3 days.

### 2.10. Formation of Burn Wounds

Standard, II–IIIa-degree burn wounds were created by exposing the shaved back skin of anaesthetised rats to a rectangular metal box with a square bottom filled with paraffin, which heats up to 1994 °F (93 °C) for 30 s. The trauma area was 4 cm^2^. Twelve hours after formation of the burn wound, it was infected with a suspension of *S. aureus*, which was chosen as the pathogenic flora because this microorganism is the most frequent causative agent of opportunistic purulent-inflammatory skin diseases. Specifically, a suspension of clinical isolate of *S. aureus* strain was applied, corresponding to 10 IU according to the turbidity standard OSO 42-28-85 of the Federal State Budgetary Institution Scientific Center for the Expertise of Medicinal Products of the Ministry of Health of Russia. Prior to the application of the bacterial suspension, perforation was carried out by applying a special mesh for better penetration of the bacterial suspension into the burnt skin surface. Then, 0.05 mL of the inoculate was dripped onto the surface and evenly distributed with a glass spatula

The rats were then randomly divided into four experimental groups:Control (medical Vaseline oil), skin (*n* = 10)Bepanthen Plus, cream for external use, skin (*n* = 10)Miramistin solution for topical application 0.01%, skin (*n* = 10)MMSC secretome-based chitosan gel (MSC-Ch-gel) (*n* = 10)

The formulations were applied 24 h later and then daily for 3 days (*n* = 5) or 7 days (*n* = 5) in each group.

After the 3 days or 7 days of treatments, the animals were euthanised by placing them in a CO_2_ chamber. The animals were then sent for dissection and pathomorphological examination to determine the rate of healing of the wound surface.

The following analyses were performed:Daily examination of the wound surface.Biochemical evaluation of total protein and creatinine in blood.Detection of *S. aureus* in the washings from the burned areas.Histological study of paraffin-embedded skin biopsies stained with haematoxylin and eosin to determine the rate of healing of the wound surface.

### 2.11. Colony-Forming Units (CFU) Counts

The contents of the wound were taken by flushing with a sterile swab immersed in 1 mL of 1% peptone water (pre-enrichment medium). The enrichment medium after flushing the swabs was then used as the initial dilution. A series of consecutive tenfold dilutions in 9 mL of 0.9% NaCl solution from 10^−1^ to 10^−7^ were prepared from a test tube with the initial dilution. Then, 0.1 mL per 2 Petri dishes with blood agar were sown from the dilutions, followed by incubation for 20 ± 4 h at t = 98.6 °F (37 °C). The calculation of the number of colony-forming units (CFU) of *Staphylococcus aureus* was made from the last dilutions where growth was noted.

### 2.12. Hystology

Skin biopsies from the wound area were fixed in 10% neutral formalin, dehydrated in alcohols of ascending concentrations and embedded in paraffin. Paraffin sections 5–7 µm thick were obtained using an SM 2000R microtome (Leica, Germany), stained with haematoxylin and eosin (H&E), and examined using a DM1000 microscope (Leica, Germany).

Semiquantitative inflammation scoring was performed by grading as follows:5—infiltration of the entire area of the wound and the border dermis.4—infiltration of 75% of the area of the wound and the border dermis.3—infiltration of 50% of the area of the wound.2—infiltration of 25% of the area of the wound.1—infiltration of 10% of the area of the wound.0—no inflammatory infiltration.

### 2.13. Statistical Analysis

The results are shown as the mean ± standard deviation (SD) or standard error of the mean (SEM). Nonparametric tests were used to compare differences among small samples (*n* ≤ 10), the Mann–Whitney U test was used to compare differences between two independent samples (control vs. experimental), and the Kruskal–Wallis test with Dunn’s post hoc analysis was used to compare differences among more than two samples. All statistical analyses were performed using GraphPad Prism software (GraphPad, La Jolla, CA, USA).

## 3. Results

### 3.1. MMSC Cultures

The MMSCs were isolated from term human placenta obtained after Caesarian section. The enzymatic approach allowed for a rapid cells isolation—they readily adhered to culture plastic on the next day after tissue digestion and reached 100% confluent monolayer within several days. At passage 3, they exhibited high proliferative activity, as numerous doubling cells were observed in culture, and were characterised by a spindle-shaped morphology (Figure 1A). Flow cytometry of passage 3 cells revealed high expression of surface markers characteristic of MMSCs population (CD105, CD90, CD44 and CD29 > 90%, CD73 > 85%) (Figure 1E–I), and negative endothelial and blood cell marker expression (CD34, CD45, CD31, CD14, CD11b, and CD19 < 1%) (Figure 1B–D). After confirming the MMSC phenotype of the culture, cells were used for the obtainment of conditioned media.

### 3.2. Protein Composition of Cell Secretome

To understand the molecular mechanism mediating the regenerative and protective properties of our secretome-based formulation, we estimated the secretome proteomic composition by liquid chromatography-tandem mass spectrometry (LC-MS/MS).

In the secretome from MMSC cultures, 281 proteins were identified. Among the identified proteins, 34.84%, 24.74%, 20.56%, and 6.62% were annotated as cytoplasmic, nuclear, extracellular, and mitochondrial, respectively (Figure 2A). Furthermore, 35.19% were annotated as proteins with catalytic activity (GO: 0003824), 26.24% were annotated as “metabolite interconversion enzyme”, 5.94% were annotated as chaperones, and 3.96% were annotated as “defence/immunity protein” (Figure 2B,C).

We also annotated the proteins for their participation in biological processes. Most proteins, 35.38% and 17.85%, are involved in “cellular process” (GO: 0009987) and “metabolic process” (GO: 0008152), respectively. In addition, 13.55% and 2.55% participate in “biological regulation” (GO: 0065007) and “immune system process” (GO: 0002376), respectively (Figure 2D).

Annotation of the secretome proteins indicated their participation in 45 pathways, including growth factor signalling pathways (FGF, VEGF, epidermal growth factor (EGF), platelet-derived growth factor (PDGF), and insulin-like growth factor (IGF)), the Wnt signalling pathway, the endothelium signalling pathway, hypoxia response, immune cell activation, angiogenesis, cytoskeletal regulation, apoptosis, and glycolysis (Figure 3).

### 3.3. Cytokine, Chemokine, and Growth Factor Levels in MMSC Secretome

Multiplex analysis of the secretome composition revealed the presence of bFGF, VEGF, MCP-1, IL-6, and IL-8 (Figure 4A–C). The secreted proteins also contained comparable levels of IL-10 and IL-17.

### 3.4. Effect of Formulations on the Biochemical Parameters of Blood Serum

No significant changes in the content of total protein and creatinine in blood serum were found after 4 and 8 days of burn wound formation in any of the animals (Figure 4D–G).

### 3.5. Antimicrobial Activity

After using the MSC-Ch-gel, Bepanthen Plus, or Miramistin, there was a complete absence of *S. aureus* in the wound (Figure 4H,I). The burn surface of the control rats was seeded with the studied microorganisms. However, there was a decrease in the contamination of the wound surface in the control group by the eighth day of observation (Figure 4I).

### 3.6. Macroscopic Examination of the Wound Surface

According to the results of the macroscopic study of the burn wounds in the rat study groups during the initial period after the burn, pronounced redness and swelling of the wound surface was noted. Exudative signs on the wound surface and in the surrounding tissues were not observed both in the experimental groups and in the control group.

On the fourth day after the application of the study drugs, no differences were found between the experimental and control animals (Figure 5A,C,E,G). The formation of a dense red-brown scab, irregular in thickness, was observed. In all studied groups, the shape of the scab did not extend beyond the contours of the thermal injury. Exudation on the wound surface and in the surrounding tissues was not observed.

On the eighth day of observation, the beginning of rejection of the wound scab was noted in the control group and the Miramistin group in some areas. The scab was dense and uneven in thickness, with consistent edges (Figure 5B,D).

In the rats treated with the MSC-Ch-gel, demarcation was also noted, the edges of the scab were uneven, and the density of the scab was less pronounced (Figure 5H).

When using Bepanthen Plus, demarcation between the scabbed burn surface and the areas of wound epithelialisation was also observed. The scab was dense and uneven in thickness; the edges were consistent, outlining the shape of the burn wound. The skin around the scab was thickened (Figure 5F).

No signs of inflammation of the tissues surrounding the wound were found in all of the study animals.

### 3.7. Histological Examination of the Wound Surface on the Fourth Day

During pathomorphological examination on the fourth day after burn creation, in all groups of animals, impairments of the skin structure were observed, characteristic of a IIIa-degree burn (Figure 6). The surface of the damaged skin areas was covered with a scab of leukocytes with inclusions of fibrin, necrotic epidermis, and dermis. In the scab, the remains of the sebaceous glands and hair follicles could be distinguished. In the connective tissue in the burn area, destruction of collagen fibres and oedema were noted; oedema of the vasculature and border dermis was also observed. Development of the epidermal ridge in all groups of animals was minimal and was characterised by hyperplastic processes in the epidermis at the border with the burn wound. In the control group, there were practically no signs of epidermisation of the wound surface (Figure 6A,B), while in the Miramistin (Figure 6C,D,H), Bepanthen Plus (Figure 6E,F,H), and MSC-Ch-gel (Figure 6G,H) groups, the epidermis grew under the scab at the edge of the wound. During treatment with Bepanthen Plus, epidermal regeneration was also noted closer to the centre of the wound in the form of a thin layer of immature epithelium. Signs of inflammation were manifested in the control and Bepanthen Plus groups. In the Miramistin group, inflammatory infiltrates were detected in small, isolated areas. In the MSC-Ch-gel-treated group, inflammatory infiltrates were absent.

### 3.8. Histological Examination of the Wound Surface on the Eighth Day

Pathomorphological examination of the structure of the skin of the rats on the eighth day after creating the burn wounds revealed signs of wound epithelialisation in all groups (Figure 7). At the same time, a scab remained on the surface of the wound. The least pronounced wound epithelialisation was found in the control group, which was due to the presence of a pronounced inflammatory reaction in the wound in response to infection (Figure 7A,B,I). In all experimental groups, epithelium grew on the wound tissue, but complete epithelialisation did not occur by the eighth day. In the groups treated with Miramistin (Figure 7C,D,I), Bepanthen Plus (Figure 7E,F,I), and the MSC-Ch-gel (Figure 7G,H,I), the newly formed epithelium exhibited signs of stratification. In the wound tissue in all groups, areas of formation of new connective tissue were noted; however, the most vascularisation for successful wound healing at the burn site and at the wound edges was observed in the group of animals treated with the MSC-Ch-gel.

In the Bepanthen Plus group, despite the high proliferative activity of the epithelium, the quality of wound epithelialisation was somewhat reduced due to its loose adherence to the wound tissue, which may be associated with ongoing inflammation in the area of the burn (Figure 7F,I). In all other experimental groups, the inflammatory infiltrate was absent or was detected in the form of small, isolated areas.

## 4. Discussion

Currently, there are many studies demonstrating the ability of MSCs derived from various tissues to stimulate wound healing and suppress inflammation [7]. However, there are comparatively fewer studies on the use of MSCs for antimicrobial therapy [17]. In particular, even less is known about the use of MSCs for the treatment of chronic bacterial infections, which are usually associated with bacterial biofilm formation and high levels of antimicrobial resistance. Recent studies showed that soluble factors secreted by MSC inhibited *Staphylococcus aureus* biofilm formation in vitro and disrupted the growth of established biofilms. Secreted factors from MSC also elicited synergistic killing of drug-resistant bacteria when combined with several major classes of antibiotics [20]. In the present work, we prepared a bio-therapeutic chitosan gel containing the secretome from placental MMSCs and estimated its antimicrobial and regenerating effects on a clinically relevant model of *S. aureus-infected burns* (II–IIIa degree) in rats. In our study, we used the clinically isolated strain of *S. aureus.* In addition to promoting skin tissue repair, we and others have found that MMSC secretome also possesses antimicrobial properties. Notably, microbiological studies have shown that MSC-Ch-gel has antimicrobial activity similar to that of Miramistin and Bepanthen Plus. Miramistin is a topical antiseptic with broad antimicrobial action, including activity against biofilms [49]. Bepanthen Plus contains the antiseptic chlorhexidine and the regenerative component dexpanthenol. *S. aureus* in the wound after the application of MSC-Ch-gel, Bepanthen Plus, and Miramistin was not observed on days 4 and 8 of the experiment. However, the burn surface not treated with drugs in the control group only showed a tendency to decrease the number of bacteria in the wound by the eighth day of observation.

We also noted almost complete epithelialisation of the wounds treated with MSC-Ch-gel, and histological examination showed signs of stratified epithelium in this group. High levels of vascularisation and angiogenesis were observed compared to those in the Bepanthen Plus and Miramistin groups. Our results are in accordance with the literature, namely that treatment with MMSCs promotes regeneration by accelerating dermal fibroblast and keratinocyte migration and proliferation, stimulating angiogenesis, and reducing scarring [13,22,47]. Of note, our proteomic findings agreed with the histological observations. We found secretome proteins involved in inflammation, angiogenesis, Wnt signalling, growth factor signalling, and the defence system, for example, which are representative of all phases of wound healing. It is known that MMSCs secrete key cytokines and growth factors that modulate inflammation and reduce cell death in the wound; they also produce immunosuppressive factors that suppress the proliferation of immune cells, including B cells, T cells, and natural killer cells in circulation [38]. MMSCs have the ability to directly influence the immunological properties of macrophages and neutrophils by secreting factors such as PGE2, IL-6, IL-8, or IFN-γ, IL-10, growth factors, and chemokines. The presence of some of these factors was detected in the placental MMSCs secretome and can promote orchestrating of all phases of skin healing.

We recognise some limitations of our research. Standardisation of secretome-based products is difficult to achieve due to the variability in cell cultures from different donors and sources. This limitation can be potentially overcome using pre-conditioned cultures or specific cell lines. Previous studies have also shown that preconditioned MSCs can improve the antimicrobial activity of certain antibiotics and secrete antimicrobial peptides such as cathelicidin LL-37, human β-defensin-2 (hBD-2), hepcidin, and lipocalin-2 (Lcn2). These effects are reproduced by the MMSC secretome [17,53,54]. Secretome regenerative capacity is also a dynamic property that depends on many factors. It was shown previously that transitioning from monolayer to non-adhesive three-dimensional culture conditions can dramatically change the secretome content and therapeutic potential [55,56]. These possibilities require further studies.

Another limitation of our study was the absence of empty chitosan gel as independent control of antimicrobial action. The aim of our study was to evaluate the complex effect of secretome-based hydrogel and the application of chitosan as an available gelling component for the rapid preparation of the drug. We supposed that the antimicrobial and anti-biofilm action of chitosan-based formulations is well known. We also recognise that the use of immunohistochemistry to assess angiogenesis and inflammation would be more compelling for a comparative assessment of the reparative effect of MSC-Ch-gel. The inflammation was nonspecific in nature and was detected on histological preparations in the form of inflammatory cellular elements or focal infiltrates diffusely located in the tissue. Polymorphonuclear leukocytes and macrophages were determined as part of inflammatory infiltrates. By day 8, there was no complete epithelialisation of the burn wound in any of the studied groups; therefore, when assessing the degree of epithelialisation of the wound, the determining feature in our study was not the length of the epithelium, but the presence or absence of epithelial stratification, i.e., the degree of its maturity. The methods used for sampling material for microbiological studies might not fully ensure the detection of microorganisms in biofilms. However, no *S. aureus* was found in histological samples. Thus, the obtained experimental data require further detailed research and clarification. We plan to expand our research to obtain a deeper understanding of the effect MMSCs secretome formulations and their applicability as a therapeutic.

However, our molecular and histological results support that the placental MSC-secretome-based chitosan gel is a promising tool to accelerate wound healing. There is reason to consider the use of placental MMSC secretome as a novel alternative treatment to augment conventional antimicrobial therapy for the treatment of chronic infections of soft tissues. We have presented a simple method to prepare the secretome-based formulation based on mixing water-soluble chitosan and conditioned medium. Considering the cost, chitosan alone, which is relatively inexpensive, could promote wound closure but it lacks the regenerative and angiogenic ability of the secretome. Obtaining and culturing MMSCs is also inexpensive, requiring no special or expensive additional agents. The secretome can be easily collected, frozen, or lyophilised and stored for an extended time. The preparation process is rapid and simple, and it can be easily implemented in clinical practice. Medical personnel can quickly prepare the formulation as needed.

## 5. Conclusions

The development of innovative regenerative products creates great opportunities to improve the treatment and quality of life of patients. Stem cell secretomes can be used to create biomimetic products with pleiotropic and multi-targeted effects. The application of such products may be able to limit the use of antibacterial and antiseptic drugs, which will reduce the risks of the spread of resistant microorganisms. However, the effects observed in our research were not optimal, and further experiments are required to improve the application of MMSC secretome-based products to burn wounds and other serious injuries. We hope that, despite the regulatory barriers and issues related to their safety, cell secretome-based products can be an effective alternative therapy in the near future.

## Figures and Tables

**Figure 1 pharmaceuticals-14-01263-f001:**
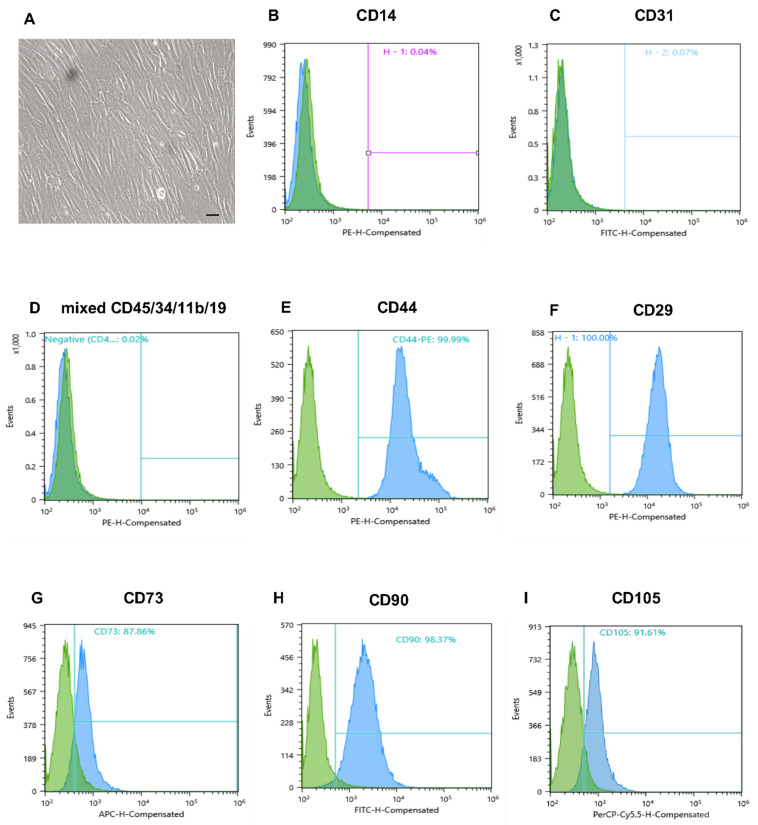
Morphology of placental MMSCs and flow cytometric analysis of specific markers of MSCs. (**A**) Third-passage MMSCs on day 3 in monolayer culture. Phase contrast microscopy, scale bars 100 μm. (**B**–**D**) Negative endothelial and blood cell marker expression. (**E**–**I**) High expression of surface marker characteristic of MMSC population. MMSC, multipotent mesenchymal stromal cell.

**Figure 2 pharmaceuticals-14-01263-f002:**
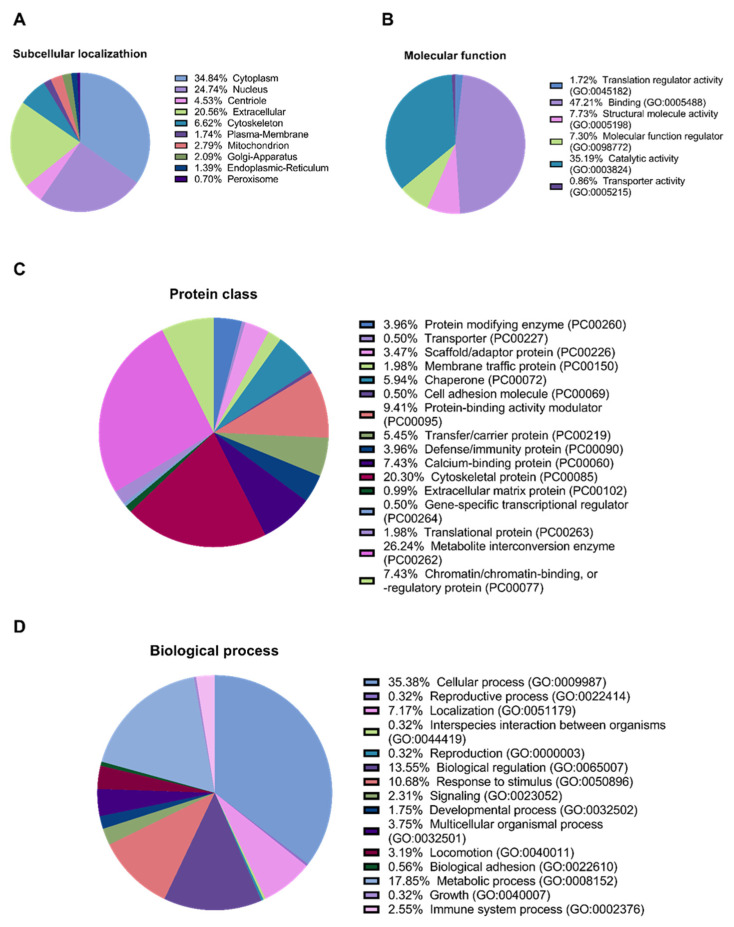
Proteomic analysis of the MMSCs secretome. (**A**) Predictor mGOASVM pie chart of subcellular localisation of proteins identified in the secretome from MMSC cell culture. (**B**) Gene ontology pie chart of molecular functions associated with proteins identified in the secretome from MMSC cell culture. (**C**) Gene ontology pie chart of protein classes associated with proteins identified in the secretome from MMSC cell culture. (**D**) Gene ontology pie chart of biological processes associated with proteins identified in the secretome from MMSC cell culture. MMSC, multipotent mesenchymal stromal cell; mGOASVM, multi-label protein subcellular localisation based on gene ontology and support vector machines.

**Figure 3 pharmaceuticals-14-01263-f003:**
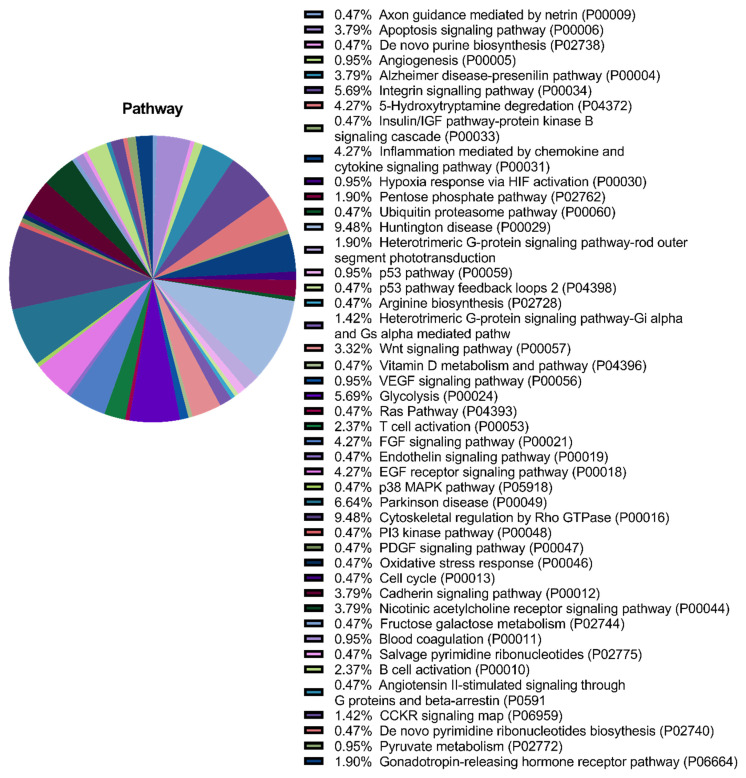
Proteomic analysis of the multipotent mesenchymal stromal cell (MMSC) secretome. Gene ontology pie chart of pathways associated with proteins identified in the secretome from MMSC cell culture.

**Figure 4 pharmaceuticals-14-01263-f004:**
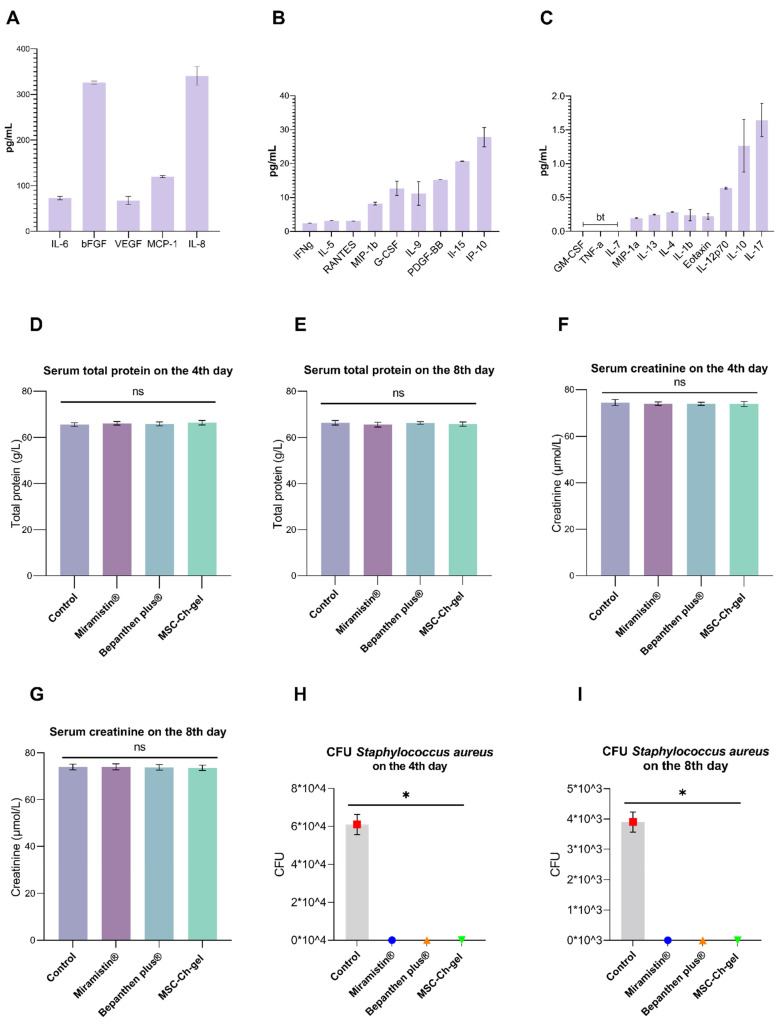
Cytokine, chemokine, and growth factor levels in the MMSC secretome and effect of cutaneous application of the investigated drugs on serum biochemical parameters of rats and on antimicrobial activity. (**A**–**C**) Plotted data are means ± standard deviation (*n* = 3 replicates). “bt” indicates a value below threshold. (**D**–**G**) Total serum protein and creatinine content 4 and 8 days after creation of the burn wound. Plotted data are the means ± standard error of the mean for each condition (*n* = 5); ns, not significant. (**H**,**I**) Number of *Staphylococcus aureus* CFU in the burn wound after cutaneous application of MSC-Ch-gel and comparison drugs, * adjusted *p* < 0.001 for Control vs. Miramistin, Bepanthen Plus, and MSC-Ch-gel groups. All statistical analyses were performed using the Kruskal–Wallis test with Dunn’s post hoc test. CFU, colony-forming unit; MSC-Ch-gel, multipotent mesenchymal stromal cell secretome-based chitosan gel.

**Figure 5 pharmaceuticals-14-01263-f005:**
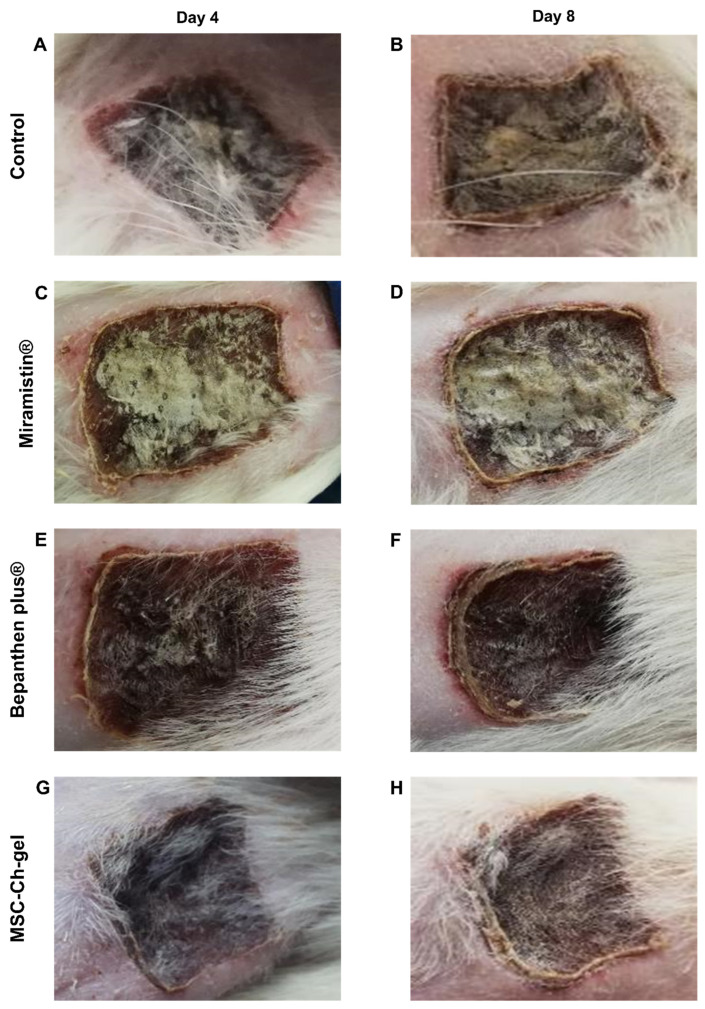
Representative visual appearance of the wound for different treatment groups on day 4 and 8. (**A**,**B**) Control. (**C**,**D**) Miramistin. (**E**,**F**) Bepanthen Plus. (**G**,**H**) MSC-Ch-gel. MSC-Ch-gel, multipotent mesenchymal stromal cell secretome-based chitosan gel.

**Figure 6 pharmaceuticals-14-01263-f006:**
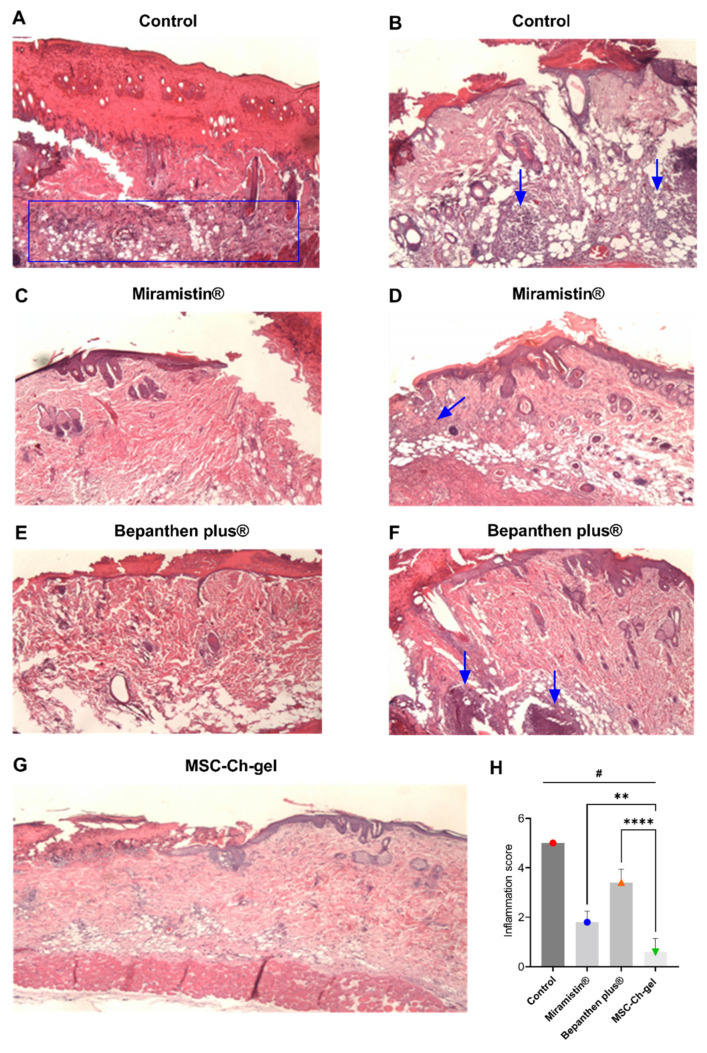
Histology of skin lesions on day 4 stained with haematoxylin and eosin, and inflammatory infiltrate score on days 4. (**A**,**B**) Control. (**C**,**D**) Miramistin. (**E**,**F**) Bepanthen Plus. (**G**) MSC-Ch-gel (magnification, ×40). The blue rectangle and arrows indicate the area of the inflammatory infiltrate. (**H**) Inflammation score. Plotted data are means ± standard deviation (*n* = 5), # adjusted *p* < 0.001 for Control vs. Miramistin, Bepanthen Plus, and MSC-Ch-gel groups, ** adjusted *p* 0.0031 for Miramistin vs. MSC-Ch-gel, **** adjusted *p* < 0.001 for Bepanthen Plus vs. MSC-Ch-gel. All statistical analyses were performed using the one-way ANOVA. To compare groups vs. Control (Dunnett’s post hoc test); to compare groups with each other (Tukey’s post hoc test). MMSC, multipotent mesenchymal stromal cell; MSC-Ch-gel, multipotent mesenchymal stromal cell secretome-based chitosan gel.

**Figure 7 pharmaceuticals-14-01263-f007:**
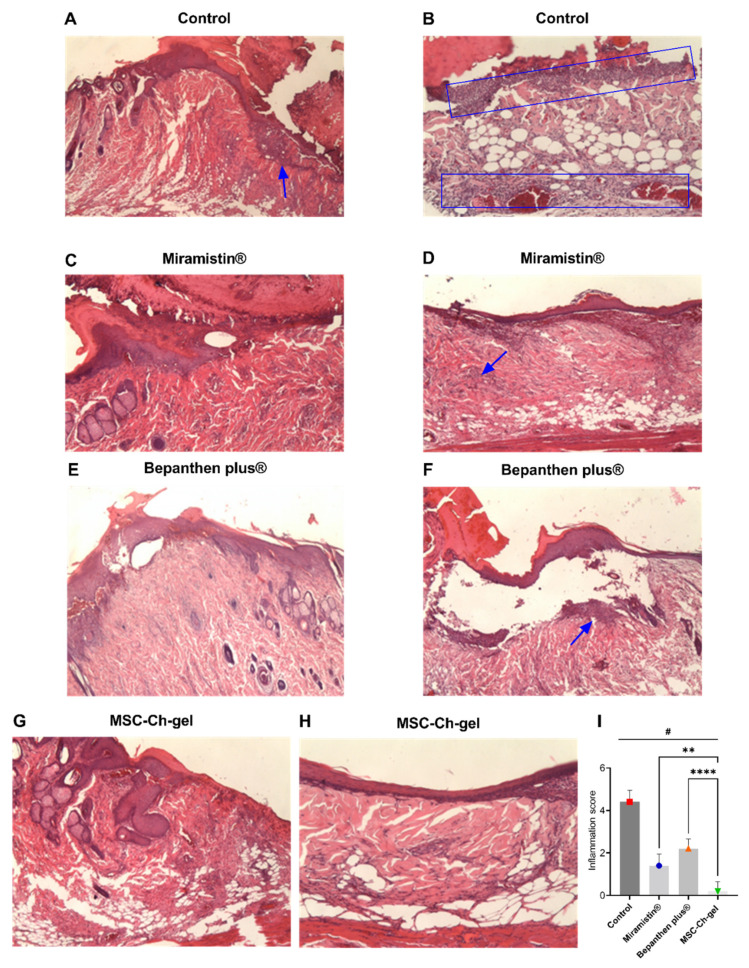
Histology of skin lesions on day 8 stained with haematoxylin and eosin and inflammatory infiltrate score on days 8. (**A**,**B**) Control (magnification, A ×25, B ×100). (**C**,**D**) Miramistin (magnification, ×40). (**E**,**F**) Bepanthen Plus (magnification, ×40). (**G**,**H**) MSC-Ch-gel (magnification, ×40, ×100). The blue rectangle and arrows indicate the area of the inflammatory infiltrate. (**I**) Inflammation score. Plotted data are means ± standard deviation (*n* = 5), # adjusted *p* < 0.001 for Control vs. Miramistin, Bepanthen Plus, and MSC-Ch-gel groups, ** adjusted *p* 0.0078 for Miramistin vs. MSC-Ch-gel, **** adjusted *p* < 0.001 for Bepanthen Plus vs. MSC-Ch-gel. All statistical analyses were performed using the one-way ANOVA. To compare groups vs Control (Dunnett’s post hoc test); to compare groups with each other (Tukey’s post hoc test). MMSC, multipotent mesenchymal stromal cell; MSC-Ch-gel, multipotent mesenchymal stromal cell secretome-based chitosan gel.

## Data Availability

Additional data can be provided upon reasonable request from the date of publication of this article within 5 years. The request should be sent to the corresponding author at cd95@mail.ru.

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
