# Peer review of "Antimicrobial and Regenerative Effects of Placental Multipotent Mesenchymal Stromal Cell Secretome-Based Chitosan Gel on Infected Burns in Rats"

_pharmaceuticals, 2021, doi:10.3390/ph14121263_

Round 1

Reviewer 1 Report

The authors made an interesting work describing the antimicrobial/regenerative effects of placental multipotent 2 mesenchymal stromal cell secretome incorporated in a chitosan-based gel on infected burns in rats. Even if the presentation is accurate I considered it should be improved by following the next suggestions:

  1. In the introduction, regarding the wound healing process and its stages these pieces of information should be more elaborated and the reference is old, from 2011, please consider citing newer paper such as: https://doi.org/10.3390/pharmaceutics12100983
  2. At line 106 the phrase: “The MMSCs isolated from term placenta adhered to culture plastic” should be more detailed, regarding the origin of the placenta…etc.
  3. I consider the order of the article impractical and illogical, the Section Materials and Methods should be the first after Introduction, and then the section Results and Discussion. Please restructure.

Author Response

Dear Reviewer,

Thank you for your interest in our research and valuable comments.

We accept all your advice and hope that the new version is much better.

Reviewer 2 Report

We would like to thank the author for the work they furnished to realize this study. If the prevention of the infection of the burned tissue is a major concern in burn care, and the use of MSC conditioned media a potential interesting therapeutics strategy, the present study contains some major methodological errors and is not suitable for publication. The proteomics analysis of the placenta MSC conditioned media is interesting, but a comprehensive analysis of the data and a comparison to the other type of MSC in the discussion is mandatory.  

  1. The MSC culture media and, therefore the MSC conditioned media contained antibiotics (100 U/mL gentamicin) that could play a role in the diminution of the infection of the wound by Staphylococcus Aureus.  The author should consider the use of media without antibiotics or the use of culture media incorporated in chitosan as control. The present study does not allow to conclude to any effect of the MSC conditioned media due to this significant bias.   
  2.  The MSC's characterization does not respond to minimal criteria for defining multipotent mesenchymal stromal cells described by the International Society for Cellular Therapy.  Flow cytometry data for placental MSCs should be presented.
  3. Analysis of LC-MS/MS data is incomplete. No statistical analysis is conducted, no pathway enrichment analysis is conducted.
  4. The description of the animal model is insufficient: the temperature of the burning device is not mentioned.  
  5. The principal result of this study is the decrease of the wound Staphylococcus aureus contamination, but the methodology used to assess this parameter is not described.  
  6. The animal study did not allow any statistical comparison. However, using complementary analysis as immunohistology to compare the vascularization or the inflammatory infiltrate of the wound bed could allow a better understanding of the potential beneficial effect of MSC conditioned media.  
  7. Histological analysis is also insufficient.
  8. No gross wound images are presented.

Author Response

(The authors gave the same response as above.)

Reviewer 3 Report

The present study examined the effectiveness of MSC secretome products delivered through a chitosan gel on burn wound healing and infection prevention.   

While an interesting approach, there are several parts that are unclear and the study is missing some controls.  Specific comments:  

Abstract: 

The abstract does not present quantitative results.  How much of a reduction in microorganisms?  how much was epithelialization enhanced?  Was epithelialization measured in this study?

Line 70, clarify that the burns were in a rat model. 

Line 94:  "good" has no specific scientific meaning.  would replace. 

In the intro, there are no studies that specifically focus on antimicrobial effects.  A few reviews are cited, but a quick literature search shows several studies on this effect and potential mechanisms.  Expanding the intro to include these would make a better rationale for this study. 

lines 149-168: multiple paragraphs discuss the same thing and standalone sentences.  Revision needed. 

line 183:  recommend revising to say that S. aureus was not detected.  These tests will not confirm that there was a "complete absence of S. aureus. 

line 189:  Is there a better word than "violations"? 

The images of histology only have qualitative descriptions and could be biased, showing only parts of the wound.  Were any attempts made to measure epithelial length?   

In line 202, a "vascular plethora" is described.   If vessels were not counted or identified in the figure, this is not valid to say. 

line 203, which specific signs of inflammation were observed in these groups? Could you identify in the figures evidence of inflammatory infiltrates?

Not sure why the graphical abstract is currently in the middle?

236-254:  Three short paragraphs can be combined into one that focuses on an overview of the study.   

One major limitation of this study is that it did not include the vehicle chitosan gel without the secretome products.   It is difficult to draw conclusions without this group.  This is not discusses in the discussion section, nor other major limitations of the study.   A major limitation is that the CFUs were only measured in "washings" from the burned area.  This method will not detect biofilm associated bacteria and could be subject to bias in the amount of volume used and retrieved. 

The discussion reads more as an introduction and is repetitive of the intro information.  This should follow the results and provide context for why results may have occurred and whther they are in agreement with other studies.   

The discussion contains several instances of result repeats.  These should be revised.   

There is minimal to no discussion of the antimicrobial results.   Have other studies observed antimicrobial effects?  why would it be antimicrobial?  

line 317 is this versene? 

The order of the materials and methods should follow the order of results.  

What is the source of the S. aureus strain? 

Why was there such a range of molecular weight for chitosan? 

What kind of heated container ? 

what media was used to suspend S. aureus prior to inoculation?  

How was the inoculum applied to the wounds?   

CFU counts are reported, but not described well in the methods nor the procedure for washing. 

Author Response

Dear Reviewer,

Thank you for your interest in our research and valuable comments.

We have found the missing of some information after English editing.

It led to misunderstanding and misconception.

We accept all your advice and hope that the new version is much better.

Round 2

Reviewer 3 Report

Comments have been addressed by authors, although some corrections would enhance the article and could be guided by editors.  

First 3 paragraphs of discussion may be better combined.  There should be no references to figures in the discussion if they are repeats of results.  Rather, the discussion should give reasons and support for the results. 

Author Response

Dear reviewer,

Thank you for your kind response and attention.

We corrected the manuscript.

With best regards,

Vasily Kudinov

With best regards,

Vasily Kudinov